# Robust Hyperbolic Learning with Curvature-Aware Optimization

**Ahmad Bdeir**
Department of Data Science
University of Hildesheim
Hildesheim, Germany
`bdeira@uni-hildesheim.de`

**Johannes Burchert**
ISMLL
University of Hildesheim
Hildesheim, Germany
`burchert@ismll.de`

**Lars Schmidt-Thieme**
ISMLL
University of Hildesheim
Hildesheim, Germany
`schmidt-thieme@ismll.de`

**Niels Landwehr**
Department of Data Science
University of Hildesheim
Hildesheim, Germany
`landwehr@uni-hildesheim.de`

## Abstract

Hyperbolic deep learning has become a growing research direction in computer vision due to the unique properties afforded by the alternate embedding space. The negative curvature and exponentially growing distance metric provide a natural framework for capturing hierarchical relationships between datapoints and allowing for finer separability between their embeddings. However, current hyperbolic learning approaches are still prone to overfitting, computationally expensive, and prone to instability, especially when attempting to learn the manifold curvature to adapt to tasks and different datasets. To address these issues, our paper presents a derivation for Riemannian AdamW that helps increase hyperbolic generalization ability. For improved stability, we introduce a novel fine-tunable hyperbolic scaling approach to constrain hyperbolic embeddings and reduce approximation errors. Using this along with our curvature-aware learning schema for Riemannian Optimizers enables the combination of curvature and non-trivialized hyperbolic parameter learning. Our approach demonstrates consistent performance improvements across Computer Vision, EEG classification, and hierarchical metric learning tasks while greatly reducing runtime. Our code is publicly available at https://github.com/inboxedshoe/Robust-Hyperbolic-Learning

## 1   Introduction

Recently, hyperbolic manifolds have gained attention in deep learning for their ability to model hierarchical and tree-like data structures efficiently. Unlike Euclidean space, hyperbolic space has negative curvature, allowing it to represent exponentially growing distances. This makes these manifolds ideal for tasks like natural language processing, graph representation, and metric learning [33]. By embedding data into hyperbolic space, models can capture complex relationships more effectively, and often with fewer parameters [11]. Hyperbolic geometry has also been applied in computer vision, where its ability to better separate embeddings in high-dimensional spaces has shown promise in improving tasks like image classification and segmentation [2, 3, 18, 40], few-shot learning [25], and feature representation [44].

These works rely on two main derivations of the hyperbolic space, the hyperboloid or Lorentz space ($\mathbb{L}$) and the Poincaré manifold ($\mathbb{P}$). Typically, the hyperboloid offers better operational stability, as

39th Conference on Neural Information Processing Systems (NeurIPS 2025).

demonstrated by Mishne et al. [29] but lacks clear definitions for basic vector operations such as addition and subtraction. To bridge this gap, recent research has focused on defining Lorentzian variant of common deep learning operations, such as the feed-forward layer [6, 8, 11], the convolutional layer [6, 8, 34], and multinomial linear regression (MLR) [3].

However, the use of these hyperbolic operations comes with challenges. Computations in hyperbolic space are more complex and expensive, and the lack of optimized CUDA implementations drastically slows down training and increases memory requirements. Additionally, optimizing hyperbolic parameters such as class prototypes or batchnorm means can be unstable, especially in low precision floating-point environments. This has required research to rely on parameter clamping techniques, which can cause non-smooth gradient updates, to remain within the accurate representation radius of the manifolds [16, 29]. The instability is only exacerbated when we incorporate the negative curvature as a learned parameter since it directly affects the embedding space. Finally, in data-scarce scenarios, the higher representational capacity makes the hyperbolic spaces more prone to overfitting, which heavily impacts their generalization ability.

This work addresses key challenges in the Lorentz model. To combat overfitting, we introduce an AdamW-based optimizer that enhances regularization and generalization in hyperbolic learning. For learning instability, we propose an optimization framework that stabilizes the learning of curvature parameters. We also present a smoother scaling function to replace weight clipping, ensuring hyperbolic vectors remain within the representational radius. To address computational complexity, we introduce an implementation trick that leverages efficient CUDA-based convolutions for hyperbolic learning. Improved stability not only boosts model performance but also enables lower-precision training, further enhancing efficiency. Our contributions are then four-fold:

1. We propose an alternative schema for Riemannian optimizers that stabilize curvature learning for hyperbolic parameters and a formulation for the Riemannian AdamW

2. We propose the use of our maximum distance rescaling function to restrain hyperbolic vectors within the representative radius of accuracy afforded by the number precision, even allowing for fp16 precision.

3. We present *LHEIR*, *HyperMAtt*, and *HCNN+* as applications of our proposed optimization scheme for the domains of hierarchical metric learning, EEG classification, and computer vision for image classification and generation, respectively.

4. We empirically show the effectiveness of our proposed methods in five domains, hierarchical metric learning, EEG classification, graph embedding, image classification, and image generation to show the effectiveness of our optimizer in different problem settings. We improve performance in all domains with a significant computational speed-up.

## 2   Related Work

**Hyperbolic Embeddings in Deep Learning**   Initially, many hyperbolic deep learning methods relied on a hybrid model architecture that utilizes Euclidean encoders and hyperbolic decoders [28]. Euclidean encoders avoid the high computational complexity of hyperbolic operations, as well as the lack of well-defined hyperbolic alternatives for Euclidean components. However, this trend has begun to shift towards fully hyperbolic models. Chen et al. [6] propose hyperbolic components for a fully connected linear layer, a graph convolution layer, and an attention layer with the square Lorentzian distance as a similarity metric. For the classification head, they learn class prototypes directly on the hyperbolic manifold and use the same distance metric as a loss estimator, following similar work in the literature [2, 19, 28]. This work was further extended to computer vision by Bdeir et al. [3] and van Spengler et al. [40] with both developing fully hyperbolic ResNets, using Riemannian batch normalization layers that rely on a learnable mean also embedded and optimized in hyperbolic space. Bdeir et al. [3] attempt to alleviate these issues by including a hybrid encoder that only applies the hyperbolic components in blocks that exhibit higher embedding hyperbolicity. Although this has led to notable performance improvements, both models suffer from upscaling issues. Attempting to apply these approaches to larger datasets or larger architectures becomes less feasible in terms of time and memory requirements. Instead, our approach places a greater focus on efficient components to leverage the beneficial hyperbolic properties of the model while minimizing the memory and computational footprint.

**Curvature Learning** Previous work in hyperbolic spaces has explored various approaches to curvature learning. In their studies, Gu et al. [14] and Giovanni et al. [13] achieve this by using a parametrization that implicitly models variable-curvature embeddings under an explicitly defined 1-curve manifold. This method enables them to simulate K-curve hyperbolic and spherical operations under constant curvature. They apply this method on mixed-curve manifold embeddings where every portion of the embedding belongs to either the Euclidean, spherical, or Poincaré manifold. Other approaches, such as the one by Kochurov et al. [20], simply set the curvature to a learnable parameter but do not account for the manifold changes while performing the optimization steps with the Riemannian optimizers. This leads to mathematical inconsistencies when updating the hyperbolic parameters, resulting in instability and accuracy degradation. Additionally, some methods, like the one by Kim et al. [19], store all manifold parameters as Euclidean vectors and project them before use. While this approach partially mitigates the issue of mismatched curvature operations, it requires repeated backpropagation through hyperbolic mappings, making it computationally expensive and more susceptible to projection errors. Other methods [3, 6, 40] use a combination of Euclidean parametrization, and direct hyperbolic parameter learning depending on the component and required precision.

## 3 Methodology

### 3.1 The Lorentz Space

The hyperbolic space is a Riemannian manifold with a constant negative sectional curvature $c < 0$. There are many conformal models of hyperbolic space but we focus our work on the hyperboloid, or Lorentz manifold. The n-dimensional Lorentz model $\mathbb{L}_K^n = (\mathcal{L}^n, \mathfrak{g}_{\boldsymbol{x}}^K)$ is defined with $\mathcal{L}^n := \{\boldsymbol{x} = [x_t, \boldsymbol{x_s}] \in \mathbb{R}^{n+1} \mid \langle \boldsymbol{x}, \boldsymbol{x} \rangle_{\mathcal{L}} = -K, \; x_t > 0\}$ implying a negative curvature of $\frac{-1}{K}$ and the Lorentzian inner product as Riemannian metric $\mathfrak{g}_{\boldsymbol{x}}^K = \langle \boldsymbol{x}, \boldsymbol{y} \rangle_{\mathcal{L}} := -x_t y_t + \boldsymbol{x}_s^T \boldsymbol{y}_s$. It then follows that $\langle \boldsymbol{x}, \boldsymbol{y} \rangle_{\mathcal{L}} \leq -K$ which is an important characteristic to note for the stability issues presented later on. The formulation $\mathcal{L}^n$ then presents the Lorentz manifold as the upper sheet of a two-sheeted hyperboloid centered at $\overline{\boldsymbol{0}}^K = [\sqrt{K}, 0, \cdots, 0]^T$. We inherit the terminology of special relativity and refer to the first dimension of a Lorentzian vector as the time component $x_t$ and the remainder of the vector as the space dimension $\boldsymbol{x_s}$. Below are the basic operations presented by the manifold with more complex operations provided in Section A in the supplementary material.

**Distance** Distance in hyperbolic space is the magnitude of the geodesic forming the shortest path between two points. Let $\boldsymbol{x}, \boldsymbol{y} \in \mathbb{L}_K^n$, then the distance between them is given by $d_{\mathbb{L}}(\boldsymbol{x}, \boldsymbol{y}) = \sqrt{K} \operatorname{acosh}\left(\frac{-\langle \boldsymbol{x}, \boldsymbol{y} \rangle_{\mathcal{L}}}{K}\right)$. We also define the square distance by Law et al. [22] as $d_{\mathbb{L}}^2(\boldsymbol{x}, \boldsymbol{y}) = \|\boldsymbol{x} - \boldsymbol{y}\|_{\mathbb{L}}^2 = -2K - 2\langle \boldsymbol{x}, \boldsymbol{y} \rangle_{\mathcal{L}}$.

**Exponential and Logarithmic Maps** Since the Lorentz space is a Riemannian manifold, it is locally Euclidean. This can best be described through the tangent space $\mathcal{T}_{\boldsymbol{x}}\mathcal{M}$, a first-order approximation of the manifold at a given point $\boldsymbol{x}$. The exponential map, $\exp_{\boldsymbol{x}}^K(\boldsymbol{z}) : \mathcal{T}_{\boldsymbol{x}}\mathbb{L}_K^n \to \mathbb{L}_K^n$ is then the operation that maps a tangent vector in $\mathcal{T}_{\boldsymbol{x}}\mathbb{L}_K^n$ onto the manifold through $\exp_{\boldsymbol{x}}^K(\boldsymbol{z}) = \cosh(\alpha)\boldsymbol{x} + \sinh(\alpha)\frac{\boldsymbol{z}}{\alpha}$, with $\alpha = \sqrt{1/K}\|\boldsymbol{z}\|_{\mathbb{L}}$, $\|\boldsymbol{z}\|_{\mathbb{L}} = \sqrt{\langle \boldsymbol{z}, \boldsymbol{z} \rangle_{\mathcal{L}}}$. The logarithmic map is the inverse of this mapping and can be described as $\log_{\boldsymbol{x}}^K(\boldsymbol{y}) = \frac{\operatorname{acosh}(\beta)}{\sqrt{\beta^2 - 1}} \cdot (\boldsymbol{y} - \beta \boldsymbol{x})$, with $\beta = -\frac{1}{K}\langle \boldsymbol{x}, \boldsymbol{y} \rangle_{\mathcal{L}}$.

### 3.2 Riemannian Optimization Schema

**Background** Many existing hyperbolic models adopt hybrid learning approaches that combine Euclidean parameterization with direct hyperbolic optimization. Our work focuses on the latter. This motivates our reliance on GeoOpt [20], a widely adopted library for Riemannian optimization based on the definitions by Bécigneul and Ganea [4], whose methodology is also reflected in related work. Here, the curvature $K$ of the hyperbolic space is set as a learnable parameter. However, empirically, we find that naively optimizing this curvature often leads to instability and performance degradation. We argue that this stems from a fundamental oversight: Riemannian optimizers update curvature without adjusting dependent hyperbolic operations, weights, or gradients, creating geometric inconsistencies.

Given a hybrid model with both Euclidean and hyperbolic parameters, we define the set of learnable parameters as $\boldsymbol{\theta} = [\boldsymbol{\theta}_{\mathbb{E}}, \boldsymbol{\theta}_{\mathbb{L}}^K, \boldsymbol{\theta}_K]$, where $\boldsymbol{\theta}_{\mathbb{E}}$ are the parameters optimized in Euclidean space, $\boldsymbol{\theta}_{\mathbb{L}}^K$ are the parameters constrained to a Riemannian manifold $\mathcal{M}$ with curvature $K$ and origin $\overline{\mathbf{0}}^K$, and $\boldsymbol{\theta}_K$ are learnable curvature parameters for the manifold. When optimizing the curvature, we refer to $\boldsymbol{\theta}_{\mathbb{L}}^{K_t}$ as the hyperbolic parameters defined on the manifold with the value of the curvature parameter at timestep $t$. It is easy to see here that $\boldsymbol{\theta}_{\mathbb{L}}^K$ is dependent on $\boldsymbol{\theta}_K$ since the hyperbolic parameters are defined by their values. Current Riemannian optimization first calculates the Euclidean gradient $\mathcal{G}$ through normal backpropagation. If the parameter we are currently updating is Euclidean ($\boldsymbol{\theta}_{\mathbb{E}}$), we perform a typical Euclidean update step. If the parameter being updated is hyperbolic ($\boldsymbol{\theta}_{\mathbb{L}}^K$), the optimizer projects the gradient onto the tangent space of the parameter $\mathcal{T}_{\boldsymbol{\theta}_{\mathbb{L}}}$ in the process egrad2rgrad described in Appendix A. Momentum vectors typically used with optimizers like Adam and SGD are also initialized and updated on the tangent space. Finally, the update step is performed using some form of retraction or exponential map.

Currently, Riemannian optimizers treat $\boldsymbol{\theta}_K$ as a Euclidean parameter. However, since the curvature defines the geometry of the manifold, changing it during training renders prior projections, gradient momentums, and $\boldsymbol{\theta}_{\mathbb{L}}^K$ parameters misaligned with the new curvature. GeoOpt attempts to mitigate this instability through an "N-stabilize step," which periodically recomputes the time components of hyperbolic parameters to ensure adherence to the manifold. However, this occurs after updates, failing to prevent invalid intermediate states during optimization.

As a concrete example, let $H$ and $H'$ be two Lorentz manifolds with curvatures $K$ and $K'$. Let $\mathbf{x}$ be a learnable model parameter that lies on $H$, meaning it satisfies $\langle \mathbf{x}, \mathbf{x} \rangle_{\mathcal{L}} = -K$. The hyperbolic distance from this point to itself, $d_{\mathcal{L}}(\mathbf{x}, \mathbf{x})$, should always be $0$. However, if the optimizer first updates the curvature to $K'$ but does not yet update the parameter $\mathbf{x}$, any subsequent operation will use the new curvature $K'$ with the old parameter coordinates. The distance calculation becomes:

$$d_{\mathcal{L}}(\mathbf{x}, \mathbf{x}) = \sqrt{K'} \cdot \text{acosh}\left( \frac{-\langle \mathbf{x}, \mathbf{x} \rangle_{\mathcal{L}}}{K'} \right) = \sqrt{K'} \cdot \text{acosh}\left( \frac{K}{K'} \right)$$

If $K < K'$, the term $\frac{K}{K'}$ is less than 1. The $\text{acosh}$ function is undefined for inputs less than 1, leading to *NaN* values and a model crash. If $K > K'$, the calculation does not crash, but the distance is no longer $0$. This incorrect distance creates invalid gradients, which destabilizes training. Previous similar works relied on rigorous training regimes with separate optimizers for the curvature, very small learning rates, and "burn-in" epochs [7, 36] which we attempt to avoid. Other solutions such as clipping may prevent undefined operations but cannot easily account for the inconsistencies in the second scenario.

**Curvature Aware Optimization**    To address this, we propose a method that groups parameters based on their respective spaces and staggers their updates. Specifically, at timestep $t$ we first isolate $\boldsymbol{\theta}_{\mathbb{L}}^{K_{t-1}}$ and update these parameters first. We then update $\boldsymbol{\theta}_{\mathbb{E}}$ followed by $\boldsymbol{\theta}_K$. At this stage, we have updated all the model parameters, but $\boldsymbol{\theta}_{\mathbb{L}}^{K_{t-1}}$ are still defined on the old curvature, so we must use a mapping function to update $\boldsymbol{\theta}_{\mathbb{L}}^{K_{t-1}} \rightarrow \boldsymbol{\theta}_{\mathbb{L}}^{K_t}$.

The N-stabilize step used by GeoOpt can be seen as a pseudo-map, but it alters the relative magnitudes of the hyperbolic parameters $\boldsymbol{\theta}_{\mathbb{L}}$ and gradients $\mathcal{G}$, as well as the directions of the momentums, which can degrade performance and lead to training instability. As such, we identify two alternate mapping techniques commonly used in the literature and compare their properties. We start with the scaling function used by Skopek et al. [36], Tabaghi et al. [38]. In their works the authors simply multiply $\boldsymbol{\theta}_{\mathbb{L}}^{K_{t-1}}$ by the scaling factor $l = \sqrt{\frac{K_t}{K_{t-1}}}$; this would automatically map all the parameters from $\mathbb{L}_{K_{t-1}}^n$ to $\mathbb{L}_{K_t}^n$ and scale all distances between points linearly by that factor. As such, it preserves the relative distances between the hyperbolic parameters.

However, the method itself presents issues during optimization and input scaling. Hyperbolic models suffer from instability and traditionally require higher precision computations to prevent invalid values. Mishne et al. [29] show that the mathematical instability is proportional to the distance between $\overline{\mathbf{0}}$ and the hyperbolic embeddings. They are also able to derive the maximum distance allowed before we begin to get undefined operations. By using the scaling factor $l$ we also scale $d_{\mathbb{L}}(\boldsymbol{x}, \overline{\mathbf{0}})$, which could then push it outside the stable range. We theorize this is why works using this method need to adhere to strict training regimes, including burn-in periods without curvature learning,

a separate optimizer for the curvature, and very low learning rates. Clipping these parameters is one solution, but larger $l$ values could make clipping too harsh, and it removes the nice property of preserved relative distances.

Additionally, there is the issue of dealing with input projections, almost all hyperbolic learning approaches assume that the Euclidean or input data exists on the tangent plane of the origin and project it onto the manifold using the exponential map. This projection preserves the norm of the Euclidean vectors as $d_{\mathbb{L}}(\boldsymbol{x}, \overline{\boldsymbol{0}})$. When the input is then projected onto the scaled manifold, the relative norms of the input and the hyperbolic class prototypes, for example, are now completely different, breaking their relationship. One would then have to find a way to scale the inputs while adhering to the maximum stability distance without clipping. This inconsistency is the same for mixed optimization settings where both Euclidean trivialized parametrization and direct hyperbolic parameter learning are used.

An alternative mapping method is presented in Fu et al. [10], Guo et al. [15] and is based on projecting the parameters onto the tangent space at the origin $\overline{\boldsymbol{0}}_{t-1}$ of the old curvature using the logarithmic map and then projecting back after the curvature update. We show in Appendix D that this method preserves the distances and angles between the hyperbolic parameters and the origin. These properties are important as they are considered proxies for hierarchy level, and embedding similarity. We also show why the tangent space at the origin is the most mathematically suitable space for this mapping. This method also mitigates the instability caused by scaling, which removes the need for a more complicated training regime and maintains symmetric parameter handling by operating analogously to trivialized parameter learning.

Given the above, our work relies on the tangent-based mapping method. We extend it to the optimization process by additionally defining the mapping of parameter gradients and momentums from $\mathcal{T}_{\boldsymbol{\theta}_{\mathbb{L}}^{\kappa_{t-1}}}$ to $\mathcal{T}_{\boldsymbol{\theta}_{\mathbb{L}}^{\kappa_t}}$. The entire mapping schema is shown in Algorithm 1. We additionally perform empirical experiments to comparing the scaling mapping and the tangent mapping in Appendix E. It is important to emphasize that our proposed optimization scheme is compatible with existing curvature learning and meta-learning methods. Rather than being an alternative, it serves as an intermediate step for updating manifold parameters during curvature changes, aimed at maintaining learning stability throughout the process.

### 3.3 Riemannian AdamW Optimizer

**Background** The AdamW optimizer was first introduced by Loshchilov and Hutter [26] and relies on an improved application of the L2-regularization factor in the base Adam optimizer. L2-regularization works on the principle that networks with smaller weights tend to have better generalization performance than equal networks with higher weight values. Adam applied the L2-regularization during the gradient update step by incorporating it in the loss. However, Loshchilov and Hutter [26] argue that this is inconsistent since the regularization effect is reduced by the magnitude of the gradient norms. Instead, they apply the weight decay directly during the parameter update step. AdamW is then shown to generalize better and lead to better convergence and has become a popular choice for many vision tasks.

Given the above, we believe AdamW is significant for hyperbolic learning. Hyperbolic spaces, with their higher representational capacity, are prone to overfitting, especially under data scarcity during training [12]. Thus, AdamW's enhanced L2-regularization could improve hyperbolic models.

**Riemannian AdamW** In the following, we derive AdamW for the Lorentz manifold and suggest its extension to the Poincaré ball. The key difference between AdamW and Riemannian Adam lies in direct weight regularization, which is challenging in Lorentz space due to the lack of an intuitive subtraction operation. One option would be following the exponential mapping and retraction typically used for the optimization step. However we propose a simpler operation that reduces the need for expensive and less stable parallel transport operations. Specifically, we re-frame parameter regularization in AdamW as a weighted centroid with the origin

$$\boldsymbol{\theta}_{t-1} - \gamma\lambda\boldsymbol{\theta}_{t-1} = (1 - \gamma\lambda)\boldsymbol{\theta}_{t-1} + \gamma\lambda\boldsymbol{O}$$

where $\gamma$ is the learning rate and $\lambda$ is the weight decay value. We can now directly translate this for hyperbolic parameters $\boldsymbol{\theta}_{\mathbb{L}}$ as $\mu_{\mathbb{L}}^{\boldsymbol{\nu}}([\boldsymbol{\theta}_{t-1}, \overline{\boldsymbol{0}}])$ where $\mu_{\mathbb{L}}^{\boldsymbol{\nu}}$ is the weighted Lorentz centroid defined in Law et al. [22] and described in Appendix A. We define the centroid weights as $\boldsymbol{\nu} = [1 - \gamma\lambda, \gamma\lambda]$.

---

**Algorithm 1** Tangent Based Manifold Mapping

---

1: **Given:**
    Hyperbolic parameters $\boldsymbol{\theta}_{\mathbb{L}}^{K}$ on the $K$-curve manifold, Parameter gradients $\mathcal{G} \in \mathcal{T}_{\boldsymbol{\theta}_{\mathbb{L}}^{K}}$
2: **function** MAP PARAMETERS
3:     **for** each $\boldsymbol{p} \in \boldsymbol{\theta}_{\mathbb{L}}$ **do**
4:         $\mathcal{G}_{\text{temp}} \leftarrow \mathcal{T}_{\boldsymbol{p} \to \overline{\mathbf{0}}_{t-1}}(\mathcal{G})$                 ▷ Parallel transport gradient to previous origin
5:         $\boldsymbol{z} \leftarrow \log_{\overline{\mathbf{0}}_{t-1}}^{K_{t-1}}(\boldsymbol{p})$                     ▷ Project parameter onto tangent space
6:         $\boldsymbol{p} \leftarrow \exp_{\mathbf{0}_{t}}^{K_{t}}(\boldsymbol{z})$                       ▷ Project back onto updated manifold
7:         $\mathcal{G} \leftarrow \mathcal{T}_{\overline{\mathbf{0}}_{t} \to \boldsymbol{p}}(\mathcal{G}_{\text{temp}})$              ▷ Transport gradient back for next update
8:     **end for**
9: **end function**

---

By removing the later gradient decay and introducing this operation as seen in Algorithm 2, we adapt AdamW for use in the Lorentz space.

---

**Algorithm 2** Riemannian Adam (RAdam) and Riemannian AdamW (RAdamW)

---

**Require:** Manifold $\mathcal{M}$, initial parameters $\boldsymbol{\theta}_{\mathbb{L}} \in \mathcal{M}$
**Require:** Learning rate $\alpha > 0$, weight decay $\lambda \geq 0$, exponential decay rates $\beta_1, \beta_2 \in [0, 1)$
**Require:** $\boldsymbol{\nu}$-weighted Lorentzian Centroid $\mu_{\mathbb{L}}^{\boldsymbol{\nu}}$
**Require:** Small constant $\epsilon > 0$, max iterations $T$
**Ensure:** Optimized parameters $p_t \in \boldsymbol{\theta}_{\mathbb{L}}$
1: Initialize moment vectors $m_0 \leftarrow 0 \in T_{p_0}\mathcal{M}$, $v_0 \leftarrow 0 \in T_{p_0}\mathcal{M}$
2: Initialize timestep $t \leftarrow 0$
3: **for** $t = 1$ **to** $T$ **do**
4:     $g_t \leftarrow \text{grad } f(p_{t-1})$ $+\lambda \cdot p_{t-1}$
5:     $\boldsymbol{\nu} \leftarrow [1 - \gamma\lambda, \gamma\lambda]$
6:     $p_{t-1} \leftarrow \mu_{\mathbb{L}}^{\boldsymbol{\nu}}([\boldsymbol{p_{t-1}}, \overline{\mathbf{0}}])$
7:     $g_t \leftarrow \text{egrad2rgrad}_{p_{t-1}}(g_t)$                      ▷ See Appendix A
8:     $m_t \leftarrow \beta_1 \cdot m_{t-1} + (1 - \beta_1) \cdot g_t$
9:     $v_t \leftarrow \beta_2 \cdot v_{t-1} + (1 - \beta_2) \cdot g_t \odot g_t$
10:    $\hat{m}_t \leftarrow m_t / (1 - \beta_1^t)$
11:    $\hat{v}_t \leftarrow v_t / (1 - \beta_2^t)$
12:    $\eta_t \leftarrow \alpha \cdot \frac{\hat{m}_t}{\sqrt{\hat{v}_t} + \epsilon}$
13:    $p_t \leftarrow \text{Retract}_{p_{t-1}}(-\eta_t)$                       ▷ See Appendix A
14:    $m_t \leftarrow \mathcal{T}_{p_{t-1} \to p_t}(m_t)$
15: **end for**
      **return** $p_t$

---

### 3.4 Maximum Distance Rescaling

**Background**    Vectors in the hyperboloid models are defined as $\boldsymbol{x} = [x_t, \boldsymbol{x_s}]^T \in \mathbb{L}_K^n$ where $x_t = \sqrt{||\boldsymbol{x}_s||^2 + K}$, $K = -1/c$ and $c$ is the manifold curvature. As such, Lorentzian projections and operations rely on the ability to accurately calculate the corresponding time component $x_t$ for the hyperbolic vectors. In their work, Mishne et al. [29] derive a maximum value for the time component $x_{t_{max}}$. Values above this push vectors off the Lorentz manifold and onto the cone defined by $x_t^2 = \sum \boldsymbol{x}_s^2$. One prominent example of instability caused by this is the inner product in $d_{\mathbb{L}}(\boldsymbol{x}, \boldsymbol{y}) = \sqrt{K}\text{acosh}\left(\frac{-\langle \boldsymbol{x}, \boldsymbol{y} \rangle_{\mathcal{L}}}{K}\right)$, where the approximations can lead to undefined mathematical values by pushing the inverse hyperbolic cosine input to less than one. Based on the above, and given a specific $K$, we can derive a maximum representational radius for the model as

$$D_{\overline{\mathbf{0}}_{\max}}^{K} = \text{acosh}\left(\frac{x_{t_{max}}}{\sqrt{K}}\right) \cdot \sqrt{K} \tag{1}$$

Under Float32 precision, and to account for values of $K < 1$ we use a limit value of $x_{t_{max}} = 2 \cdot 10^3$. When projected onto the tangent space of the origin, this translates to $\|\log_{\overline{\mathbf{0}}} \mathbf{x}\|_{\mathbb{E}} = D_{\overline{\mathbf{0}}_{max}} = 9.1$. Vectors outside this radius lead to instability and performance degradation due to inaccurate approximation. This problem is only exacerbated as the dimensionality of the hyperbolic vector increases. Higher dimensional vectors tend to have larger norms which limits hyperbolic models' abilities to scale up.

To constrain hyperbolic vectors within a specified maximum distance, either a normalization function or a parameter clipping method is required. Parameter clipping can be challenging as it may lead to information loss and introduce non-smooth gradients. On the other hand, common normalization functions like tanh and the sigmoid function tend to saturate quickly, limiting their effectiveness as seen in the sigmoid implementation by Chen et al. [6].

**Flexible Scaling Function**   To address these issues, we introduce a modified scaling function, designed to provide finer control over both the maximum values and the slope of the curve. A visualization of this function is provided in Figure 2, and the formulation is presented below:

$$\mathbf{y}_{\text{rescaled}} = \frac{\mathbf{y}}{\|\mathbf{y}\|} \cdot m \cdot \tanh\left( \|\mathbf{y}\| \cdot \frac{\text{atanh}(0.99)}{s \cdot m} \right) \tag{2}$$

where $\mathbf{y} \in \mathbb{R}^d$, $m$ is our desired maximum value, and $s$ controls the slope of the curve. We now have a maximum distance value to adhere to and a flexible normalizing function. To apply this to the hyperbolic embeddings, we suggest performing the scaling on the tangent plane of the origin. However, this is an expensive operation to perform repeatedly, as such we derive in Section C the equivalent factorized form for the scaling of the space values:

$$\mathbf{x}_{s_{\text{rescaled}}} = \mathbf{x}_s \times \frac{e^{\frac{D(\mathbf{x},\overline{\mathbf{0}})^K_{\text{rescaled}}}{\sqrt{K}}} - e^{\frac{-D(\mathbf{x},\overline{\mathbf{0}})^K_{\text{rescaled}}}{\sqrt{K}}}}{e^{\frac{D(\mathbf{x},\overline{\mathbf{0}})^K}{\sqrt{K}}} - e^{\frac{-D(\mathbf{x},\overline{\mathbf{0}})^K}{\sqrt{K}}}} \tag{3}$$

where $D(\mathbf{x}, \overline{\mathbf{0}})^K_{\text{rescaled}}$ are the distances obtained by plugging $D(\mathbf{x}, \overline{\mathbf{0}})$ and $D^K_{\overline{\mathbf{0}}_{max}}$ in Equation (2).

## 4   Experiments

In order to empirically prove the effectiveness of our proposed solutions, we apply them to metric learning to test low precision learning, EEG classification for generalization ability in data hungry scenarios, and image classification and generation tasks for component efficiency. We also apply the curvature learning for most scenarios to study the new optimizer scheme and possible benefits. We include more detailed ablations and experiments on graph embeddings in the Appendix E.

### 4.1   Hierarchical Metric Learning Problem

**Problem Setting and Reference Model**   In their paper Kim et al. [19] take on the problem of hierarchical clustering using an unsupervised hyperbolic loss regularizer they name HIER. This method relies on the use of hierarchical proxies as learnable ancestors of the embedded data points in hyperbolic space. In the following experiment, we extend HIER to the Lorentz model (LHIER) and compare against the results provided by Kim et al. [19].

**Moving to Lorentz Space**   To adapt the HIER model to the hyperboloid, we first replace the Euclidean layer norm and linear layer with a Lorentzian norm and linear layer [3]. We then modify the HIER loss by replacing the Poincaré distance with the Lorentzian distance and optimizing Lorentzian hierarchical proxy parameters directly on the manifold. We use our new optimization schema and apply distance rescaling before layer norm and after the linear layer.

**Experimental Goals**   Kim et al. [19] employ the Euclidean AdamW optimizer with Euclidean parameterizations of hyperbolic proxies. Their experiments use FP16 precision. As the setting is already hyperbolic, significant gains from transitioning to Lorentz space are unlikely. We use this setup to evaluate our components' ability to learn curvature in low-precision, unstable environments.

Table 1: Performance of metric learning methods on the four datasets as provided by [19]. † indicates models using larger input images. Network architectures are abbreviated as, R–ResNet50 [41].

| Methods | Arch. | CUB | | Cars | | SOP | |
|---|---|---|---|---|---|---|---|
| | | R@1 | R@2 | R@1 | R@2 | R@1 | R@10 |
| *CNN Backbone* | | | | | | | |
| NSoftmax [45] | $R^{512}$ | 61.3 | 73.9 | 84.2 | 90.4 | 78.2 | 90.6 |
| †ProxyNCA++ [39] | $R^{512}$ | 69.0 | 79.8 | 86.5 | 92.5 | 80.7 | 92.0 |
| Hyp [9] | $R^{512}$ | 65.5 | 76.2 | 81.9 | 88.8 | 79.9 | 91.5 |
| HIER [19] | $R^{512}$ | 70.1 | 79.4 | 88.2 | 93.0 | 80.2 | 91.5 |
| LHIER | $R^{512}$ | **73.4** | **82.4** | **90.0** | **94.0** | **81.9** | **93.1** |
| Increase in % | | 4.71 | 3.78 | 2.04 | 1.08 | 1.49 | 1.20 |

Any improvements likely stem from better curvature adaptation to the data and task. We follow the experimental setup in Kim et al. [19].

**Results**  As shown in Table 1, LHIER learns curvature without stability issues. Our approach improves model performance, with recall@1 gains ranging from 1.49% to 4.71%. Additional ablation experiments with different optimizers and configurations are detailed in Table 7.

## 4.2 EEG Classification Problem

**Problem Setting**  Given EEG recordings labeled into distinct categories, the goal is to classify new recordings based on signal patterns. Each recording is a time series $X \in \mathbb{R}^{C \times T}$, where $C$ is the number of channels and $T$ is the time length. Each label is $y \in \{1, \dots, K\}$. Given $N$ labeled recordings $((X_1, y_1), \dots, (X_N, y_N))$ from an unknown distribution $p$, the task is to train a model $\hat{y}$ that maps EEG signals $X$ to their correct class.

**Reference Model**  In their work, Pan et al. [32] introduces Matt, a Riemannian model based on the SPD manifold. The model processes data through convolutional denoisers, extracts embedding covariances, and embeds them using a Riemannian attention mechanism before projecting the outputs back into Euclidean space for classification via a linear layer. Our proposed model, HyperMatt, extends this framework by projecting the resulting covariances onto the hyperboloid instead and processing them with a hyperbolic attention layer [6]. The outputs are then classified directly on the manifold using the MLR [3].

**Experimental Goals**  Hyperbolic spaces are prone to overfitting, especially in EEG classification, where models are trained per subject with limited sessions, leading to data scarcity. The proposed RAdamW optimizer theoretically improves regularization, ensuring smoother convergence. We apply our method to the datasets and the evaluation criteria in Pan et al. [32] to test this.

**Results**  In Table 2 we show the performance of *HyperMAtt* compared to the current state-of-the-art baselines, where the results were aggregated from [5, 32]. The performance for *HyperMAtt* is measured across 10 runs and we report the mean and standard deviation. Here, our new optimizer achieves state-of-the-art results for SSVEP and ERN, the two comparatively smaller datasets, which are more prone to overfitting. We also show the improved performance of our RAdamW optimizer vs the existing RAdam by training the model using both and comparing the results in Table 8. We also achieve these results in significantly less time/epoch compared to MAtt. Specifically for MI 0.077s/epoch vs 3.4s/epoch, for SSVEP 0.081s/epoch vs 4s/epoch, and for ERN 0.061s/epoch vs 1.9s/epoch, resulting in an average speed-up by a factor of 42.

## 4.3 Standard Image Classification Problem

**Problem Setting and Model**  In their work, Bdeir et al. [3] propose a fully hyperbolic and hybrid encoder RessNet. To adapt our methods for both models, we introduce the distance rescaling function after every convolution in the encoder. Additionally, we attempt to improve the performance by performing a parametrization trick on existing CUDA convolutions to ensure hyperbolic outputs.

Table 2: Performance comparison for the EEG datasets MI, SSVEP, and ERN. We report the average accuracy for MI and SSVEP and the AUC for ERN. The best result is highlighted in bold.

| Models | MI | SSVEP | ERN |
|---|---|---|---|
| ShallowConvNet[35] | 61.84±6.39 | 56.93±6.97 | 71.86±2.64 |
| EEGNet[23] | 57.43±6.25 | 53.72±7.23 | 74.28±2.47 |
| SCCNet[42] | 71.95±5.05 | 62.11±7.70 | 70.93±2.31 |
| EEG-TCNet[17] | 67.09±4.66 | 55.45±7.66 | 77.05±2.46 |
| TCNet-Fusion[31] | 56.52±3.07 | 45.00±6.45 | 70.46±2.94 |
| FBCNet[27] | 71.45±4.45 | 53.09±5.67 | 60.47±3.06 |
| MBEEGSE[1] | 64.58±6.07 | 56.45±7.27 | 75.46±2.34 |
| Inception[5] | 62.85±3.21 | 62.71±2.95 | 73.55±5.08 |
| MAtt[32] | **74.71**±5.01 | 65.50±8.20 | 76.01±2.28 |
| HyperMAtt | 74.13±3.09 | **68.12**±2.63 | **77.98**±1.60 |
| Increase in % | -0.78 | 4.01 | 2.59 |

This is done by parametrizing the convolution weight as a rotation operation before passing the input, and then applying a boost operation afterwards. The end result is then equivalent to the convolution operation by Bdeir et al. [3] while mitigating computational complexity. Specific implementation details can be found in Section B. We denote our Hybrid and Fully hyperbolic models as HECNN+ and HCNN+ respectively.

**Experimental Goals** In their paper, [3] notice lower performance in the fully hyperbolic models compared to the hybrid models and attribute this to instability in training. Additionally, they cite crashes and model divergence when learning the curvature instead of setting it to constant. This would then be an ideal setting to test the optimization schema and rescaling function and their impact on performance and stability.

**Results** In the ResNet-50 experiments in Table 3, HECNN+ significantly outperforms both the Euclidean model and the base hybrid model across both datasets, demonstrating the positiv effect of curvature learning. We also see a $\sim 48\%$ reduction in memory usage and $\sim 66\%$ reduction in runtime. We attribute this improvement to efficient closed-source CUDA convolution operations we can now leverage. One other benefit that we find from learning the curvature is quicker convergence, where the model is able to reach convergence in 130 epochs vs the 200 epochs required by a static curve model. ResNet-18 experiments show similar findings and can be found in Appendix E.

Table 3: Performance and runtime Analysis for ResNet-50 models. We report classification accuracy (%) and the best performance is highlighted in bold (higher is better). All experiments were conducted on a single NVIDIA RTX 4090 GPU and an AMD EPYC 7543 CPU.

| | CIFAR-100 ($\delta_{rel} = 0.23$) | Tiny-ImageNet ($\delta_{rel} = 0.20$) | VRAM For Cifar100 | $t_{epoch}$ For Cifar100 |
|---|---|---|---|---|
| Euclid | 78.52 | 66.23 | 4.5GB | 30s |
| HECNN | 79.83 | 66.30 | 15.6GB | 300s |
| HECNN+ | **80.86** | **67.18** | 8.1GB | 100s |

## 4.4 VAE Image Generation

**Experimental Setup** We reproduce the experimental setup from [3] and re-implement the fully hyperbolic VAE using our new efficient convolution and transpose convolution layers. We also use curvature learning with our adjusted Riemannian SGD learning scheme.

**Results** Our fully hyperbolic VAE implementation outperforms on both datasets, as shown in Table 4, while using 2.5x less memory and training 3x faster. This highlights the effectiveness of our curvature learning process and efficient model components.

Table 4: Reconstruction and generation FID of manifold VAEs across five runs (lower is better).

| | CIFAR-100 | | CelebA | |
|---|---|---|---|---|
| | Rec. FID | Gen. FID | Rec. FID | Gen. FID |
| Euclid | $63.81_{\pm 0.47}$ | $103.54_{\pm 0.84}$ | $54.80_{\pm 0.29}$ | $79.25_{\pm 0.89}$ |
| Hybrid ($\mathbb{P}$) | $62.64_{\pm 0.43}$ | $98.19_{\pm 0.57}$ | $54.62_{\pm 0.61}$ | $81.30_{\pm 0.56}$ |
| Hybrid ($\mathbb{L}$) | $62.14_{\pm 0.35}$ | $98.34_{\pm 0.62}$ | $54.64_{\pm 0.34}$ | $82.78_{\pm 0.93}$ |
| HCNN ($\mathbb{L}$) | $\underline{61.44}_{\pm 0.64}$ | $\underline{100.27}_{\pm 0.84}$ | $\underline{54.17}_{\pm 0.66}$ | $\underline{78.11}_{\pm 0.95}$ |
| HCNN+ ($\mathbb{L}$) | $\mathbf{57.69}_{\pm 0.52}$ | $\mathbf{98.14}_{\pm 0.44}$ | $\mathbf{52.73}_{\pm 0.27}$ | $\mathbf{77.98}_{\pm 0.32}$ |
| Decrease in % | 6.10 | 2.12 | 2.66 | 0.17 |

## 5 Conclusion

In this work, we propose a robust curvature-aware optimization framework for hyperbolic deep learning, addressing challenges in stability, computational efficiency, and overfitting. By introducing Riemannian AdamW, a novel distance rescaling function, and leveraging efficient CUDA implementations, our approach achieved performance improvements in hierarchical metric learning, EEG classification, and image classification tasks, while reducing computational costs, highlighting the practicality of our methods for large-scale applications. These findings emphasize the importance of proper curvature adaptation in hyperbolic learning, paving the way for future research in optimizing hyperbolic embeddings across diverse fields. Further work should be done however to address the limitations on theoretical reasoning and efficiency presented in Appendix F.

## 6 Acknowledgements

Ahmad Bdeir was funded by the European Union's Horizon 2020 research and innovation programme under the SustInAfrica grant agreement No 861924, as well as Horizon Europe Project BioMonitor4CAP grant agreement No 101081964.

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

# A  Operations in hyperbolic geometry

**Parallel Transport**  A parallel transport operation $\mathrm{PT}^K_{\boldsymbol{x}\to\boldsymbol{y}}(\boldsymbol{v})$ describes the mapping of a vector on the manifold $\boldsymbol{v}$ from the tangent space of $\boldsymbol{x}\in\mathbb{L}$ to the tangent space of $\boldsymbol{y}\in\mathbb{L}$. This operation is given as $\mathrm{PT}^K_{\boldsymbol{x}\to\boldsymbol{y}}(\boldsymbol{v}) = \boldsymbol{v} + \frac{\langle\boldsymbol{y},\boldsymbol{v}\rangle_{\mathcal{L}}}{K - \langle\boldsymbol{x},\boldsymbol{y}\rangle_{\mathcal{L}}}(\boldsymbol{x} + \boldsymbol{y})$.

**Lorentzian Centroid [22]**  Also denoted as $\boldsymbol{\mu}_{\mathbb{L}}$, is the weighted centroid between points on the manifold based on the Lorentzian square distance. Given the weights $\boldsymbol{\nu}$, $\boldsymbol{\mu} = \frac{\sum_{i=1}^{m} \nu_i \boldsymbol{x}_i}{\sqrt{1/K}\left|\|\sum_{i=1}^{m} \nu_i \boldsymbol{x}_i\|_{\mathcal{L}}\right|}$.

**Optimization Operations**  egrad2rgrad converts a Euclidean gradient (computed in the ambient space) to a Riemannian gradient by projecting it onto the tangent of the corresponding hyperbolic parameter. For the Lorentz manifold this is then:

$$\mathrm{egrad2rgrad}(\mathbf{x}, v) = v + \frac{\langle v, \mathbf{x}\rangle_{\mathcal{L}}}{K}\mathbf{x}.$$

The retraction maps the updated hyperbolic vector from the tangent space back onto the manifold. In the case of the hyperboloid this is the proposed exponential mapping equation.

**Lorentz Transformations**  In the Lorentz model, linear transformations preserving the structure of spacetime are termed Lorentz transformations. A matrix $\mathbf{A}^{(n+1)\times(n+1)}$ is defined as a Lorentz transformation if it provides a linear mapping from $\mathbb{R}^{n+1}$ to $\mathbb{R}^{n+1}$ that preserves the inner product, i.e., $\langle\mathbf{A}\boldsymbol{x}, \mathbf{A}\boldsymbol{y}\rangle_{\mathcal{L}} = \langle\boldsymbol{x}, \boldsymbol{y}\rangle_{\mathcal{L}}$ for all $\boldsymbol{x}, \boldsymbol{y} \in \mathbb{R}^{n+1}$. The collection of these matrices forms an orthogonal group, denoted $\boldsymbol{O}(1, n)$, which is commonly referred to as the Lorentz group.

In this model, we restrict attention to transformations that preserve the positive time orientation, operating within the upper sheet of the two-sheeted hyperboloid. Accordingly, the transformations we consider lie within the positive Lorentz group, denoted $\boldsymbol{O}^+(1, n) = \mathbf{A} \in \boldsymbol{O}(1, n) : a_{11} > 0$, ensuring preservation of the time component sign $x_t$ for any $\boldsymbol{x} \in \mathbb{L}^n_K$. Specifically, in this context, Lorentz transformations satisfy the relation

$$\boldsymbol{O}^+(1, n) = \mathbf{A} \in \mathbb{R}^{(n+1)\times(n+1)} | \forall \boldsymbol{x} \in \mathbb{L}^n_K : \langle\mathbf{A}\boldsymbol{x}, \mathbf{A}\boldsymbol{x}\rangle_{\mathcal{L}} = -\frac{1}{K}, (\mathbf{A}\boldsymbol{x})_0 > 0). \tag{4}$$

Each Lorentz transformation can be decomposed via polar decomposition into a Lorentz rotation and a Lorentz boost, expressed as $\mathbf{A} = \mathbf{RB}$ [30]. The rotation matrix $\mathbf{R}$ is designed to rotate points around the time axis and is defined as

$$\mathbf{R} = \begin{bmatrix} 1 & \mathbf{0}^T \\ \mathbf{0} & \tilde{\mathbf{R}} \end{bmatrix}, \tag{5}$$

where $\mathbf{0}$ represents a zero vector, $\tilde{\mathbf{R}}$ satisfies $\tilde{\mathbf{R}}^T \tilde{\mathbf{R}} = \mathbf{I}$, and $\det(\tilde{\mathbf{R}}) = 1$. This structure shows that Lorentz rotations on the upper hyperboloid sheet belong to a special orthogonal subgroup, $\boldsymbol{SO}^+(1, n)$, which preserves orientation, with $\tilde{\mathbf{R}} \in \boldsymbol{SO}(n)$.

In contrast, the Lorentz boost applies shifts along spatial axes given a velocity vector $\boldsymbol{v} \in \mathbb{R}^n$ with $\|\boldsymbol{v}\| < 1$, without altering the time axis.

$$\mathbf{B} = \begin{bmatrix} \gamma & -\gamma\boldsymbol{v}^T \\ -\gamma\boldsymbol{v} & \mathbf{I} + \frac{\gamma^2}{1+\gamma}\boldsymbol{v}\boldsymbol{v}^T \end{bmatrix}, \tag{6}$$

with $\gamma = \frac{1}{\sqrt{1-\|\boldsymbol{v}\|^2}}$. However, this can also be any operation that scales the norms of the space values without changing the vector orientation.

# B  Convolution Trick

To address the computational requirements issue, we adopt an alternative definition of the Lorentz Linear layer from Dai et al. [8], which decomposes the transformation into a Lorentz boost and a

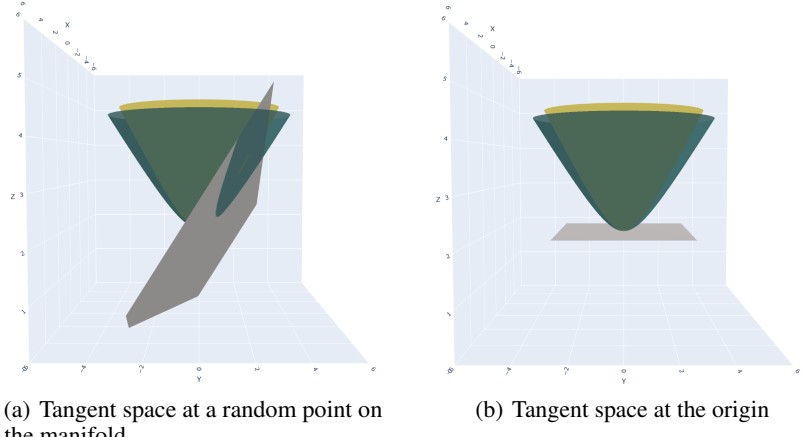

(a) Tangent space at a random point on the manifold

(b) Tangent space at the origin

Figure 1: Tangent planes of a hyperboloid with curvature -1 relative to another hyperboloid with curvature -0.7. Tangential properties between manifolds are better respected at the origin where tangents remain parallel.

Lorentz rotation. Using this definition, we replace the matrix multiplication employed by Bdeir et al. [3] for the spatial dimensions and time component projection with a learned rotation operation and a Lorentz boost. Additionally, we can achieve the rotation operation using a parameterization of the convolution weights while still relying on the CUDA convolution implementations, significantly improving computational efficiency.

To apply this concept to the convolution operation, the convolution weights, after unfolding, must form a rotation matrix. We define the dimensions of this matrix as $n = (channels_{in} \cdot kernel_{width} \cdot kernel_{height})$ and $n' = channels_{out}$ respectively. We then use either rotation operation presented above to a norm-preserving transformation $\boldsymbol{z} = \boldsymbol{W^T}\boldsymbol{x} \cdot \frac{\|\boldsymbol{x}\|}{\|\boldsymbol{W^T}\boldsymbol{x}\|}$ where $\boldsymbol{W} \in \mathbb{R}^{(n' \cdot n)}$. This formulation allows us to utilize existing efficient implementations of the convolution operation by directly parameterizing the kernel weights before passing them into the convolutional layer. Finally, we formalize the new Lorentz Convolution as:

$$\boldsymbol{out} = \text{LorentzBoost}(\text{DistanceRescaling}(\text{RotationConvolution}(\boldsymbol{x}))) \tag{7}$$

where TanhRescaling is the operation described in Eq.3 and RotationConvolution is a normal convolution parameterized through the procedure in Algorithm 3 where Transform is the norm-preserving transformation above.

---

**Algorithm 3** Lorentz Convolution Parameterization

---

1: $\boldsymbol{W} \in \mathbb{R}^{C_{\text{in}}, C_{\text{out}}, K_{\text{width}}, K_{\text{length}}}$
2: **function** ADAPTWEIGHT
3:     **if** $K_{\text{width}} \cdot K_{\text{length}} \cdot C_{\text{in}} \leq C_{\text{out}}$ **then**
4:         $\boldsymbol{W} \leftarrow \text{reshape}(\boldsymbol{W}, K_{\text{width}} \cdot K_{\text{length}} \cdot C_{\text{in}}, C_{\text{out}})$
5:         $\hat{\boldsymbol{W}}_{\text{core}} \leftarrow \text{Transform}(\boldsymbol{W})$
6:         $\boldsymbol{W} \leftarrow \text{reshape}(\hat{\boldsymbol{W}}, C_{\text{in}}, C_{\text{out}}, K_{\text{width}}, K_{\text{length}})$
7:     **end if**
8:     **return** $\boldsymbol{W}$
9: **end function**

---

## C   Scaling Lorentzian Vectors

**Tanh Scaling**   We show the output of the tanh scaling function in Figure 2. By changing the transformation parameters we are able to fine-tune the maximum output and the slope to match our desired function response.

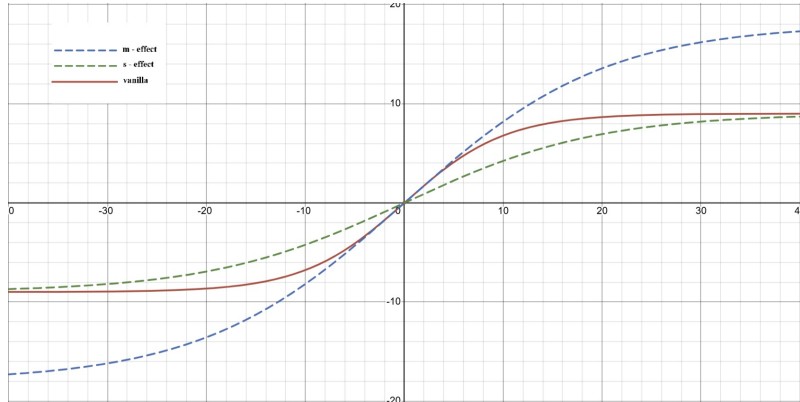

Figure 2: The output of the proposed flexible tanh function. Here the maximum value m is set to 9.1 in the vanilla version with an alternate value of m=18 and the slope s is set to 2.6 with an alternate value of 3.5

**Hyperbolic Scaling**    We isolate the transformation of the $\exp_{\bar{\mathbf{0}}}^{K}(y)$ operation on the space values of $y$ as:

$$\boldsymbol{x}_s = \sqrt{K} \times sinh(\frac{\|\boldsymbol{y}\|_{\mathbb{L}}}{\sqrt{K}})\frac{\boldsymbol{y}}{\|\boldsymbol{y}\|_{\mathbb{L}}} \tag{8}$$

where $\boldsymbol{y} \in \mathbb{R}^d = \log_{\bar{\mathbf{0}}}^{K}(\boldsymbol{x})$. However, at the tangent plane of the origin, the first element $\boldsymbol{y}_0$ becomes 0. As such $\|\boldsymbol{y}\|_{\mathbb{L}} = \|\boldsymbol{y}\|_{\mathbb{E}} = \sum_{i=2}^{d} \boldsymbol{y}_i^2$. This gives us:

$$\boldsymbol{x}_s = \sqrt{K} \times sinh(\frac{\|\boldsymbol{y}\|_{\mathbb{E}}}{\sqrt{K}})\frac{\boldsymbol{y}}{\|\boldsymbol{y}\|_{\mathbb{E}}} \tag{9}$$

We can now scale the norm of the Euclidean vector $\boldsymbol{y}$ bay a value $a$ and find the equivalent value for the hyperbolic space elements:

$$a_{\mathbb{L}} = \frac{\boldsymbol{x}_{s_{rescaled}}}{\boldsymbol{x}_s} = \frac{sinh(\frac{a \times \|\boldsymbol{y}\|_{\mathbb{E}}}{\sqrt{K}})}{sinh(\frac{\|\boldsymbol{y}\|_{\mathbb{E}}}{\sqrt{K}})} = \frac{e^{\frac{a \times \|\boldsymbol{y}\|_{\mathbb{E}}}{\sqrt{K}}} - e^{\frac{-a \times \|\boldsymbol{y}\|_{\mathbb{E}}}{\sqrt{K}}}}{e^{\frac{\|\boldsymbol{y}\|_{\mathbb{E}}}{\sqrt{K}}} - e^{\frac{-\|\boldsymbol{y}\|_{\mathbb{E}}}{\sqrt{K}}}} \tag{10}$$

Additionally, we know that the hyperbolic distance from the origin of the manifold to any point is equal to the norm of the projected vector onto the tangent plane. Supposing that we want $a \times D(\boldsymbol{x}, \bar{\mathbf{0}})^K = D(\boldsymbol{x}, \bar{\mathbf{0}})^K_{rescaled}$, we get the final equation:

$$\boldsymbol{x}_{s_{rescaled}} = \boldsymbol{x}_s \times \frac{e^{\frac{D(\boldsymbol{x}, \bar{\mathbf{0}})^K_{rescaled}}{\sqrt{K}}} - e^{\frac{-D(\boldsymbol{x}, \bar{\mathbf{0}})^K_{rescaled}}{\sqrt{K}}}}{e^{\frac{D(\boldsymbol{x}, \bar{\mathbf{0}})^K}{\sqrt{K}}} - e^{\frac{-D(\boldsymbol{x}, \bar{\mathbf{0}})^K}{\sqrt{K}}}} \tag{11}$$

## D   Proofs

In the following, we show that the distances and angles between the origin and the hyperbolic points are preserved. This is relatively trivial but we add it for sake of clarity. For this discussion, we work in the Lorentz manifold and repeat the definitions here for easier referencing.

$\mathcal{L}^n := \{\mathbf{x} = [x_t, \mathbf{x_s}] \in \mathbb{R}^{n+1} \mid \langle \mathbf{x}, \mathbf{x} \rangle_{\mathcal{L}} = -K, \ x_t > 0\}, < \mathbf{x}, \mathbf{y} >_{\mathcal{L}} = -x_t y_t + \mathbf{x}_s^T \mathbf{y}_s$ and the origin is $O = [O_0, \mathbf{O}_s] = [\sqrt{K}, 0, \dots, 0]$.

We also define two hyperboloids $H$ and $H'$ with curvatures $-K$ and $-K'$ respectively. Let $x, y$ be two points on $H$ and $v, w \in T_O H$ be their logarithmic map onto the tangent space at the origin of $H$.

**Proof of distance preservation**   We can show that distances to the origin are preserved because $d_{\mathbb{L}}(\mathbf{x}, \mathbf{O})$ collapses to $||v||_{euclid}$. This gives $d_{\mathbb{L}}(\mathbf{x}, \mathbf{O}) = ||v||_{euclid} = d_{\mathbb{L}}(\mathbf{x}', \mathbf{O}')$.

**Proof of angle preservation**   We can similarly show that angles w.r.t to the origin are preserved. The angle $\theta$ between $x$ and $y$ at $O$ is actually computed via the Euclidean inner product in $T_O H$ using

$$\cos\theta \frac{<v, w>}{||v||||w||}$$

As such the angle w.r.t. origins also does not change when moving between manifolds of different curvature using this method since $v$ and $w$ don't change.

# E   Additional Experiments

## E.1   Image Classification

**Problem Setting and Reference Model**   In their work, Bdeir et al. [3] proposed a fully hyperbolic 2D convolutional layer by breaking down the convolution operation into a window-unfolding step followed by a modified version of the Lorentz Linear Layer from Chen et al. [6]. This approach ensured that the convolution outputs remained on the hyperboloid. However, the manual patch creation combined with matrix multiplication made the computation extremely expensive, as it prevented the use of highly optimized CUDA implementations for convolutions. As such, the authors proposed a two versions of their hyperbolic ResNet classifier. A fully hyperbolic model with all Lorentz ResNet blocks (HCNN) and a hybrid encoder model with alternating Euclidean and hyperbolic blocks (HECNN).

Table 5: Performance and runtime Analysis for the ResNet-18 models. We report classification accuracy (%) and estimate the mean and standard deviation from five runs. The best performance is highlighted in bold (higher is better).

| | CIFAR-100 $(\delta_{rel} = 0.23)$ | Tiny-ImageNet $(\delta_{rel} = 0.20)$ | VRAM For Cifar100 | $t_{epoch}$ For Cifar100 |
|---|---|---|---|---|
| Euclidean | $77.72_{\pm 0.15}$ | $65.19_{\pm 0.12}$ | 1.2GB | 12s |
| Hybrid Poincaré [16] | $77.19_{\pm 0.50}$ | $64.93_{\pm 0.38}$ | - | - |
| Hybrid Lorentz [3] | $78.03_{\pm 0.21}$ | $65.63_{\pm 0.10}$ | - | - |
| Poincaré ResNet [40] | $76.60_{\pm 0.32}$ | $62.01_{\pm 0.56}$ | - | - |
| HECNN [3]($\mathbb{L}$) | $78.76_{\pm 0.24}$ | $65.96_{\pm 0.18}$ | 4.3GB | 100s |
| HECNN+ (ours) | $78.80_{\pm 0.12}$ | $65.98_{\pm 0.11}$ | 3GB | 80s |
| HCNN [3] ($\mathbb{L}$) | $78.07_{\pm 0.17}$ | $65.71_{\pm 0.13}$ | 10GB | 175s |
| HCNN+ (ours) | $\mathbf{78.81_{\pm 0.19}}$ | $\mathbf{66.12_{\pm 0.14}}$ | 5GB | 140s |

**Resnet-18 Experiments**   Table 5 shows that the new models are able to remain stable while learning the curvature. Additionally, we see significant performance improvements between HCNN and HCNN+, where our proposed architecture now matches the performance of the hybrid encoders. We hypothesize that the improved scaling function helps mitigate the previous performance inconsistencies. However, the performance difference between the hybrid models is not significant, we hypothesize this is due to the alternating architecture which could limit the effect and instability of hyperbolic components. Finally, both models manage to maintain performance while reducing the memory footprint and runtime by approximately $25 - 50\%$ and $18 - 25\%$ respectively.

**Ablation Experiments**   In table 6, we run ablation experiments to verify the effectiveness of our individual components. Specifically, the default case refers to the current model with the tanh rescaling function, learnable curvature, and our proposed optimization scheme. In the setting "fixed curve" we use a non-learnable curvature $K = 1$, and keep the tanh rescaling. The use of the new optimization schema here has no effect since

Table 6: Resnet-50 Ablations on Cifar-100.

| | CIFAR-100 |
|---|---|
| HCNN+ - Default | **80.86** |
| HCNN+ - fixed curve | 79.6 |
| HCNN+ - no scaling | 80.13 |
| HCNN+ - no optim scheme | $NaaN$ |

Table 7: Ablation on components for Metric Learning

| Methods | Arch. | CUB | | Cars | | SOP | |
|---|---|---|---|---|---|---|---|
| | | R@1 | R@2 | R@1 | R@2 | R@1 | R@10 |
| *CNN Backbone* | | | | | | | |
| LHIER - RSGD | $R^{512}$ | 64.2 | 72.1 | 73.9 | 81.4 | 67.8 | 73.9 |
| LHIER - RAdam | $R^{512}$ | 70.1 | 79.8 | 87.6 | 92.0 | 79.8 | 90.3 |
| LHIER - Fixed Curvature | $R^{512}$ | 71.2 | 80.8 | 88.7 | 91.8 | 79.1 | 89.7 |
| LHIER - No Optim Scheme | $R^{512}$ | 65.8 | 72.0 | - | - | - | - |
| LHIER - Default | $R^{512}$ | **73.4** | **82.4** | **90.0** | **94.0** | **81.9** | **93.1** |

curvature is not learnable which means the staggered
updates and parameter projections are not needed. In
the setting "no scaling" we continue to learn the curvature with the new optimization schema but
do not use the tanh rescaling. And in the setting "no optim scheme", we learn the curvature and
use the scaling but do not use the new optimization schema. As we can see, the best results are
achieved when all the architectural components are included. In the case of attempting to learn
the curvature without the proposed optimizer schema, the model breaks completely down due to
excessive numerical inaccuracies.

## E.2 Metric Learning

**Problem Setting and Reference Model**  In their paper [19] rely on a hybrid hyperbolic architecture
for modeling hierarchical relationships in image datasets. They include a new hyperbolic metric
learning loss dubbed HIER which takes into consideration the intrinsic hierarchy in the data. Given
a triplet of points $xi, xj, xk$ where $x_i$ and $x_j$ are determined to be related by a reciprocal nearest
neighbor measure, and $x_k$ is an unrelated point, the HIER loss is calculated as

$$
\begin{aligned}
\mathcal{L}_{\text{HIER}}(t) = & [D_B(x_i, \rho_{ij}) - D_B(x_i, \rho_{ijk}) + \delta]_+ \\
& + [D_B(x_j, \rho_{ij}) - D_B(x_j, \rho_{ijk}) + \delta]_+ \\
& + [D_B(x_k, \rho_{ijk}) - D_B(x_k, \rho_{ij}) + \delta]_+,
\end{aligned}
\tag{12}
$$

where $D_B$ denotes the hyperbolic distance on the Poincaré ball, and $\rho_{ij}$ is the most likely least
common ancestor of points $x_i$ and $x_j$. This encourages a smaller hyperbolic distance between $x_i$,
$x_j$, and $\rho_{ij}$, and a larger distance with $\rho_{ijk}$. The opposite signal is then applied in the case of $x_k$,
the irrelevant data point. Kim et al. [19] show substantial performance uplifts for the HIER loss
when applied to a variety of network architectures. We rely on four main datasets: CUB-200-2011
(CUB)[43], Cars-196 (Cars)[21], Stanford Online Product (SOP)[37], and In-shop Clothes Retrieval
(InShop)[24]. Performance is measured using Recall@k, the fraction of queries with at least one
relevant sample in their k-nearest neighbors. All model backbones are pre-trained on ImageNet for
fair comparisons.

**Ablation Experiments**  In table 7, we verify the effectiveness of our RAdamW optimizer, along
with the additional components proposed. Specifically, "default" refers to the current LHIER model
with learnable curvature and the new optimization scheme trained using RAdamW. The remaining
settings are the same as default but differ in the mentioned component e.g. "RSGD" is trained with
RSGD instead of RAdamW.

## E.3 EEG Classification

**Ablation Experiments**  We also train the model with the original Adam Optimizer and include the
results in Table 8. This clearly shows the improvement provided by our RAdamW optimizer over
standard RAdam.

## E.4 Graph Learning

**Problem Setting**  We follow the experimental setup described in Chen et al. [6] for the node
classification problem in four popular graph embedding datasets. We choose the node classification

Table 8: Performance comparison for the EEG datasets MI, SSVEP, and ERN. We report the average accuracy for MI and SSVEP and the AUC for ERN. The best result is highlighted in bold.

| Models | MI | SSVEP | ERN |
|---|---|---|---|
| **HyperMAtt + RAdam** | 68,53±1.29 | 62.42±2.62 | 74.98±5.84 |
| **HyperMAtt + RAdamW** | **74.12**±2.91 | **68.10**±2.41 | **78.01**±1.30 |

task since the task decoder used by the fully hyperbolic graph convolution network (GCN) [6] relies on class prototypes learned directly on the hyperboloid. Additionally, the model is already Lorentz-based, this means we do not need to modify it, and we can test our components directly in a simple problem setting. Table 9 summarizes the training conditions for the ablations performed. Here, a checked AdamW refers to the use of the Euclidean AdamW for the trivialized model (hyperbolic parameters learned on the tangent space of the origin) and our proposed hyperbolic RAdamW for the others. We keep the same hyperparameters defined in Chen et al. [6] for all training settings. Additionally, for the scalemap-based method we scale the input data to match the new norms of the manifold before projecting.

**Results** From table 9 we can see that that the model using Learnable curvature, our RAdamW and the tangent-based scaling achieves the highest performance. It should also be noted that the performance values for the the HyboNet paper are extracted directly from the paper [6], we were able to reproduce all results except the Disease dataset where we could only achieve a score of 91% using their setting.

Table 9: Results on the node classification task using GCNs.

| | Trivialized | AdamW | Learnable K | Scale Map | Disease | Airport | PubMed | Cora |
|---|---|---|---|---|---|---|---|---|
| HyboNet | - | - | - | - | **96** | 90.9 | 78 | 80.2 |
| H1 | - | - | ✓ | - | 94.09 | 92.56 | 76.60 | 80.80 |
| H2 | - | ✓ | - | - | 92.89 | 93.25 | 77.23 | 81.5 |
| H3 | - | ✓ | ✓ | - | 94.82 | **93.6** | **78.3** | **82.1** |
| H4 | ✓ | ✓ | ✓ | - | 91.94 | 91.3 | 77.10 | 80.3 |
| H5 | - | ✓ | ✓ | ✓ | 94.88 | 93.55 | 78 | 81.8 |

# F    Limitations

The proposed work still suffers greatly from the computational complexity introduced by hyperbolic operations. The effect of approximations for common operations such as the exponential map and the logarithmic map should be studied to reduce these issues. Additionally, alternatives for moving between manifolds should be researched and correlated with the task at hand and the loss being used. The different properties for the different alternatives could be better leveraged depending on the desired component outputs.

