# OpenReview forum: "Robust Hyperbolic Learning with Curvature-Aware Optimization"
_NeurIPS.cc/2025/Conference — NeurIPS 2025 poster_

### Official Review · Reviewer_7vca · 2025-06-28

**Clarity:** 3
**Significance:** 3
**Originality:** 3
**Rating:** 4
**Confidence:** 4

**Summary:**

This paper proposes a curvature-aware optimization framework for hyperbolic deep learning, introducing a Riemannian AdamW optimizer, distance rescaling, and stable curvature learning. The method enhances training stability, generalization, and efficiency for hyperbolic models across diverse tasks. Extensive experiments show consistent performance gains and reduced computational costs in metric learning, EEG classification, and image generation.

**Questions:**

- The paper highlights improved stability when learning curvature. Please provide concrete cases where baseline methods diverge or crash under curvature learning, and what specific instability symptoms are observed.

- In section 4.3, could the authors clarify whether these CUDA layers will be released publicly? If not, can the efficiency gains be replicated using standard PyTorch ops or approximated with open libraries?

- In Section 3.3, how sensitive is the performance to the weight decay and learning rate values in the Riemannian AdamW optimizer? Have these been tuned per task?

- In Section 3.4, “flexible scaling function”, how does the choice of slope “s” affect training stability or convergence? Was this fixed across all experiments?

**Ethical Concerns:**

["NO or VERY MINOR ethics concerns only"]

**Final Justification:**

I thank the authors for their rebuttal, which successfully addressed all of my initial questions. After carefully reading the other reviewers' feedback, I agree that the paper makes a valuable contribution by providing a robust and efficient optimization framework for hyperbolic networks. Therefore, I am maintaining my initial positive rating.

**Limitations:**

Yes

**Quality:**

3

**Strengths And Weaknesses:**

Strengths:
- The proposed method appears theoretically sound, and addressing important challenges such as instability and overfitting during curvature learning.

- The paper introduces a Riemannian AdamW variant that adapts a widely used optimizer to non-Euclidean space, improving regularization.

- The proposed distance rescaling function helps to keep embeddings within numerically stable regions, and the tangent-space-based parameter remapping strategy prevents inconsistency during curvature updates.

- The method demonstrates
   + strong empirical performance across tasks like hierarchical metric learning, EEG classification, and image recognition, with consistent improvements over existing baselines.
   + performance gains under limited data, highlighting good generalization.

- The method reduces memory usage, and converges faster, and improves training stability.


Weaknesses:
- While the method claims to improve training stability under curvature learning, it is not always clear what baseline methods crash or fail, and under what exact conditions.

- In section 4.3, lines 322-323, the reported efficiency gains rely on “closed-source CUDA convolution operations,” which raises concerns about reproducibility and whether others can replicate these results.

- The notation and math in Sections 3.1-3.4 are dense and may be hard to follow without more visual aids or worked examples. This limits clarity.

- In Section 4.4, line 332, the paper reports 2.5x lower memory usage and 3x faster training for the VAE, but does not provide any system details, hardware configuration, or concrete memory/runtime numbers to support this claim.

- The paper proposes several components (optimizer, scaling, mapping), but the contributions of each are spread across different tasks. It’s difficult to assess how much each component helps individually.

---

> ### Author Rebuttal · Authors · 2025-07-31
>
> Thank you for your interest in the work!
>
> ### Q1. "Please provide concrete cases where baseline methods diverge or crash under curvature learning, and what specific instability symptoms are observed."
>
> Instability arises when an optimizer updates the manifold's curvature (K) without correctly mapping the parameters on that manifold to the new geometry. This creates "geometric inconsistencies" where fundamental mathematical properties of the space are violated during updates, leading to crashes or incorrect gradients.
>
> Here is a concrete example:
>
> For easier reference we'll repeat some definitions here $\mathcal{L}^n := [\mathbf{x}=[x_t, \mathbf{x_s}]\in \mathbb{R}^{n+1}|<\mathbf{x}, \mathbf{x}>_{\mathcal{L}}= -K,x_t>0]$
>
> where $<\mathbf{x},\mathbf{y}>_{\mathcal{L}}= -x_ty_t+\mathbf{x}_s^T\mathbf{y}_s$
>
> Let $H$ and $H'$ be two Lorentz manifolds with curvatures $K$ and $K'$. Let $x$ be a learnable model parameter that lies on $H$, meaning it satisfies $\langle x, x \rangle_{\mathcal{L}} = -K$. The hyperbolic distance from this point to itself, $d_{\mathbb{L}}(x,x)$, should always be $0$.
>
> If the optimizer first updates the curvature to $K'$ but does not yet update the parameter $x$, any subsequent operation will use the new curvature $K'$ with the old parameter coordinates. The distance calculation becomes:
> $d_{\mathbb{L}}(x,x) = \sqrt{K'} \cdot \text{acosh}\left(\frac{-\langle x, x \rangle_{\mathcal{L}}}{K'}\right) = \sqrt{K'} \cdot \text{acosh}\left(\frac{K}{K'}\right)$
> * If $K < K'$, the term $\frac{K}{K'}$ is less than $1$. The $\text{acosh}$ function is undefined for inputs less than $1$, leading to \textit{NaN} values and a model crash.
>
> * If $K > K'$, the calculation does not crash, but the distance is no longer $0$. This incorrect distance creates invalid gradients, which destabilizes training and requires the very strict training regimes (e.g., small learning rates, long warm-ups) that our method seeks to avoid.
>
> This is why previous similar works rely on rigorous training regimes with separate optimizers for the curvature, very small learning rates, and "burn-in" epochs [5,6,7]. Other solutions such as clipping may prevent undefined operations but cannot easily account for the inconsistencies in the second scenario.
>
> &nbsp;
>
> ### Q2. "...efficiency gains rely on 'closed-source CUDA convolution operations,' which raises concerns about reproducibility... will be released publicly?"
>
> This is a misunderstanding, apologies for the confusion. We do not develop any novel closed-source CUDA operations. Instead, our method provides a novel reparameterization that allows us to leverage the highly optimized, standard PyTorch convolution layers (which use closed-source CUDA implementations under the hood) for hyperbolic operations.
>
> Normally, these standard layers cannot be used as they do not preserve the manifold's geometry. Our work describes the hyperbolic convolution as a sequence of a rotation and a boost. By reparameterizing the PyTorch convolution weights to perform the rotation, we can use the efficient native operation and then apply a standard boost operation. This method is detailed in Appendix B. It still does not achieve the same speeds as the default pytorch convolutions since the parametrization is non-trivial and requires matrix projections and reshaping but it is much faster and more memory efficient than the naive convolution operations used in previous work [1] and is fully reproducible with standard libraries.
>
> &nbsp;
>
> ### Q3. "...does not provide any system details, hardware configuration, or concrete memory/runtime numbers to support this claim."
>
> Thank you for pointing this out, this should indeed have been given in the paper. All experiments were conducted on a single NVIDIA RTX 4090 GPU with 24GB of VRAM and an AMD EPYC 7543 CPU. We provide the concrete memory and runtime numbers for the VAE experiment below. These results are consistent with the efficiency gains for ResNet-50 reported in Table 3 of the main paper.
>
> | Model Type          | Runtime (s/epoch) | VRAM (GB) |
> | ------------------- | ----------------- | --------- |
> | With Bdeir Conv [1] | 58                | 13.2      |
> | With Our Conv       | 19                | 5.48      |
>
> &nbsp;
>
> ### Q4. "the contributions of each are spread across different tasks. It's difficult to assess how much each component helps individually."
>
> We were unfortunately unable to add these ablations studies to the  main paper due to the page limit, however, the appendix (specifically tables 6, 7, 8, and 9) have extensive experiments with and without each of the included components in all of the experimental settings. If you believe the experiments could benefit from any particular additional ablations, please let us know and we can try to add them in as well.
>
> &nbsp;
>
> ### Q5. "...how sensitive is the performance to the weight decay and learning rate values in the Riemannian AdamW optimizer? Have these been tuned per task?"
> For most experiments, we found the optimizer to be robust and simply used the same hyperparameters as the original Euclidean baseline models. The one exception was the EEG classification task, which is known to be sensitive to hyperparameters due to high noise and limited per-subject data. For this task, we performed a small grid search over 3 learning rates and 3 weight decay values to find the optimal configuration, following prior work.
>
> &nbsp;
>
> ### Q6. "...how does the choice of slope 's' affect training stability or convergence? Was this fixed across all experiments?"
>
> The slope s was fixed to a value of 3 for all experiments across all tasks and datasets. While we observed in preliminary experiments that performance can degrade with very low (s<1) or very high (s>6) slope values, the model was not overly sensitive to the exact value within a reasonable range. We chose to fix the value rather than tune it per task to avoid introducing an additional hyperparameter to the optimization process and to demonstrate the general robustness of the approach.
>
> &nbsp;
>
> [1] Bdeir, Ahmad, Kristian Schwethelm, and Niels Landwehr. "Fully hyperbolic convolutional neural networks for computer vision." ICLR 2024

---

> > ### Comment · Reviewer_7vca · 2025-08-06
> >
> > I appreciate the authors' responses. The technical explanations about instability due to curvature updates are convincing and address my concern. The CUDA and hardware questions are clarified, and ablation/optimizer sensitivity responses are satisfactory. I encourage the authors to incorporate these clarifications in the final version, especially regarding the system details and references to appendix tables, to improve clarity.

---

> > > ### Author Response · Authors · 2025-08-06
> > >
> > > We thank you for the thorough review and the suggested updates. We'll update the manuscript to incorporate the hardware details and better clarify the instabilities, CUDA process, ablations, and sensitivities. We're very grateful for the positive assessment.

---

### Official Review · Reviewer_Es5F · 2025-07-01

**Clarity:** 3
**Significance:** 2
**Originality:** 2
**Rating:** 4
**Confidence:** 3

**Summary:**

This paper proposes techniques to improve stability and efficiency in the context of deep hyperbolic neural networks (HNNs). In particular, it addresses challenges of HNNs based on the Lorentz model: overfitting, learning instability, and computational complexity. To this aim, the authors present the following solutions: (1) Extending the optimization method in Fu et al. [9], Guo et al. [14] for dealing with geometric inconsistencies during the update step of hyperbolic parameters (2) Extending AdamW for optimization on Lorentz space (3) A function to rescale points on the Lorentz space that copes with instability an performance degradation due to inaccurate approximation. The authors provide experimental evaluations on hierarchical metric learning, EEG classification, image classification and generation that show the relevance of the proposed techniques.

**Questions:**

- In Section 3.3, does the issue only apply to hyperbolic space or to other Riemannian manifolds as well ? Can
the proposed optimizer be effective for other spaces ?

- Could the authors provide more details on HyperMatt: how covariance matrices are mapped onto the hyperboloid and
how the hyperbolic attention layer [6] is used in the next step ?

- Results on EEG signal classification are obtained over 3 runs which does not seem to be the same as the experimental setting
used by the baseline MAtt (average over 10 runs). Since standard derivations of most methods are high, it is important to see how the proposed method perform for several runs.

- While results on image generation show improvement w.r.t. existing hybrid and fully hyperbolic models, they seem to be largely
outperformed by state-of-the-art methods on the consider task (e.g. diffusion models). I am wondering if those experiments are worth
investigating for demonstrating the interest of the proposed techniques. How the methods in Table 4 perform w.r.t. different resolutions of input images (e.g. $8 \times 8$  on CIFAR-100) ?

**Ethical Concerns:**

["NO or VERY MINOR ethics concerns only"]

**Final Justification:**

This paper proposes techniques to address the problems of overfitting, learning instability, and computational complexity when training HNNs with the Lorentz model. The practical benefits of the proposed methods have been demonstrated in four different applications. While in some cases, the proposed methods do not show state-of-the-art results compared to existing works, I think these methods are promising to advance the relevant literature. During the rebuttal, the authors have addressed my concerns appropriately in their answers as I have confirmed in my discussion. I did not update my original score because it is already above 3, and I think the impact of the proposed methods to a broader range of applications is somewhat limited (due to the performance issue mentioned above).

**Limitations:**

Yes

**Paper Formatting Concerns:**

I did not notice any major formatting issues in this paper

**Quality:**

2

**Strengths And Weaknesses:**

Strengths:
- The paper deals with some key issues in learning on hyperbolic space (in particular Lorentz space).
- Overall, the paper is well-structured and easy to follow
- The proposed methods improve state-of-the-art HNNs in terms of accuracy and computational efficiency

Weaknesses:
- Please see the questions below

---

> ### Author Rebuttal · Authors · 2025-07-31
>
> Thank you for your thorough review and constructive feedback.
>
>
> ### Q1: "issue only apply to hyperbolic space or to other Riemannian manifolds as well ?"
>
> We show a direct example of the instability from optimization in the rebuttal to reviewer K73U, and while this example is specific to the Lorentz manifold, it can be extended to other manifolds. For example, we define two Poincare manifolds $P$ and $P'$ with radii $K$ and $K'$ respectively. Let $x$ be a point on $P$. The distance from x to the origin is defined as $d_\kappa(\mathbf{x}, \mathbf{0}) = K \cdot \operatorname{atanh}\left(\frac{|\mathbf{x}|}{K}\right)$, normally this is well-defined because for any $\mathbf{x}$ in $P$, $|\mathbf{x}| < K$. However, if we now update $K \rightarrow K'$ and $K' < K$, the upper bound of $\frac{|\mathbf{x}|}{K}$ is no longer 1 and we get undefined values. For $K' > K$ we get geometric inconsistencies where the distances between points are distorted non-linearly.
>
> Even in spherical manifolds, the operations rely on the property $|\mathbf{x}| = \sqrt{K}$ which becomes invalid if we update K without updating the vector and use the new operations anyways. As such, our learning scheme should be able to apply to any Riemannian manifold that has a strict condition on its vectors w.r.t its curvature.
>
> &nbsp;
>
> ### Q2. "Could the authors provide more details on HyperMatt"
>
> We first project the covariance matrix into the ambient Euclidean space through the use of the logmap. In order to prevent duplicate values, we take the upper half of the matrix and flatten it, and then we project it onto the hyperboloid using the expmap at the origin. Technically, there is no isometric mapping between the SPD manifold and the Lorentz manifold which is why we leverage the ambient Euclidean space as a middle point for the mapping. We have also tried to use treat the SPD values as Euclidean and directly calculate a time component for them but that yielded worse results.
>
>
> &nbsp;
>
>
> ### Q3. "Results on EEG signal classification are obtained over 3 runs which does not seem to be the same as the experimental setting used by the baseline MAtt (average over 10 runs)."
>
> This is an excellent point. We have rerun the EEG classification experiments for 10 runs to align with the baseline's evaluation protocol. We report the updated mean and standard deviation below and have included the original MAtt results for convenience. While the standard deviation has changed slightly, our method's strong performance holds, and we will update the main paper with these new results.
>
> | Model     | MI               | SSVEP            | ERN              |
> | --------- | ---------------- | ---------------- | ---------------- |
> | MAtt      | $74.71 \pm 5.01$ | $65.50 \pm 8.2$  | $76.01 \pm 2.28$ |
> | HyperMAtt | $74.13 \pm 3.09$ | $68.12 \pm 2.63$ | $77.98 \pm 1.6$  |
>
> &nbsp;
>
> ### Q4. "...they seem to be largely outperformed by state-of-the-art methods on the considered task (e.g. diffusion models)."
>
> This is a fair observation. While our work provides significant computational efficiency improvements for hyperbolic models, translating the largest state-of-the-art generative models (e.g., diffusion models) is still very challenging computationally. Our goal was to demonstrate the effectiveness of our components in established hyperbolic VAE architectures. We agree that scaling these methods to SOTA generative frameworks such as diffusion models is a very interesting future direction, but we believe the work required would merit a separate, dedicated study.

---

> > ### Comment · Reviewer_Es5F · 2025-08-05
> > **Thank you very much for your rebuttal**
> >
> > Thank you for considering all my questions and the detailed answers. My concerns have been addressed and I have no further questions for now.
> >
> > Please update the paper with the new results based on 10 runs for fair comparison with previous work. While the main strength of the paper seems to lie in its practical aspect and there are not many of theoretical contributions, I think it is still valuable for improving existing hyperbolic neural networks. For this reason, I remain positive about the paper.

---

> > > ### Author Response · Authors · 2025-08-06
> > >
> > > Thank you for your valuable feedback and continued support. We’re glad your concerns have been addressed, and we agree that the fair comparison is very important.  We will incorporate the results into the updated manuscript.

---

### Official Review · Reviewer_k73U · 2025-07-02

**Clarity:** 4
**Significance:** 2
**Originality:** 2
**Rating:** 4
**Confidence:** 3

**Summary:**

The paper proposes an alternative to Riemannian optimizers, adapted to Hyperbolic Neural Networks on the Lorentz manifold model of hyperbolic space. The optimizer is based on AdamW optimizer, here adapted to computations in the Lorentz model, together with a so-called "flexible scaling function" replacing the cutoff or normalization usually done at fixed radius in the manifold's chart centered at the origin, which has to be introduced due to numerical precision reasons. Together these technical tricks allow for better performance on a series of experiments on HNN tasks across varied architectures.

**Questions:**

1) Line 39-40: About the sentence "The instability is only exacerbated [...]"  can you provide a quantitative/empirical proof of this?

2) Line 128-129: "fundamental oversight: Riemannian optimizers update curvatures without adjusting dependent hyperbolic operations, weights, or gradients, creating geometric inconsistencies" -- what do you meas by "geometric inconsistencies" and why do they matter at all? Please give some concrete examples.

3) Line 244: You say that the flexible scaling function is useful and allows to control the difficulties mentioned in the paragraph of lines 239-243. So can you be more specific or concrete in comparing with cutoff/normalization, to prove the principles that you say the new scaling function helps with? So for example, do you have measures of gradients going to zero under normalization, and not with your new function, or measures of gradient non-smoothness under clipping, compared to your method? This (in the form of an ablation study/comparison on the experiments you already have) would help in order to justify better this method.

**Ethical Concerns:**

["NO or VERY MINOR ethics concerns only"]

**Final Justification:**

The authors addressed minor questions that I have, and confirmed my understanding of the paper. I therefore did not change my scores, and am comfortable with them.

**Limitations:**

I can't imagine potential negative societal impacts, so I think this point is mute for this paper.

**Paper Formatting Concerns:**

no concerns.

**Quality:**

3

**Strengths And Weaknesses:**

The strength of the paper is shown by the numerical experiments, showing better performance than previous optimizers.

The weakness is that this is a technical paper, which some reviewers (not me) would consider like an incremental improvement to previous works. I believe that such improvement, be it incremental or not, is important for advancing the field of HNNs.

---

> ### Author Rebuttal · Authors · 2025-07-31
>
> Thank you for your detailed feedback and insightful questions! We hope we can address any concerns below.
>
> &nbsp;
>
> ### Question 1: "Line 39-40: About the sentence 'The instability is only exacerbated [...]' can you provide a quantitative/empirical proof of this?"
> ### Question 2: "Line 128-129: '...what do you mean by 'geometric inconsistencies' and why do they matter at all? Please give some concrete examples."
>
> We address these two questions, which are both related to instability, together.
>
> Instability arises when an optimizer updates the manifold's curvature (K) without correctly mapping the parameters on that manifold to the new geometry. This creates "geometric inconsistencies" where fundamental mathematical properties of the space are violated during updates, leading to crashes or incorrect gradients.
>
> Here is a concrete example:
>
> For easier reference we'll repeat some definitions here $\mathcal{L}^n := [\mathbf{x}=[x_t, \mathbf{x_s}]\in \mathbb{R}^{n+1}|<\mathbf{x}, \mathbf{x}>_{\mathcal{L}}= -K,x_t>0]$
>
> where $<\mathbf{x},\mathbf{y}>_{\mathcal{L}}= -x_ty_t+\mathbf{x}_s^T\mathbf{y}_s$
>
> Let $H$ and $H'$ be two Lorentz manifolds with curvatures $K$ and $K'$. Let $x$ be a learnable model parameter that lies on $H$, meaning it satisfies $\langle x, x \rangle_{\mathcal{L}} = -K$. The hyperbolic distance from this point to itself, $d_{\mathbb{L}}(x,x)$, should always be $0$.
>
> If the optimizer first updates the curvature to $K'$ but does not yet update the parameter $x$, any subsequent operation will use the new curvature $K'$ with the old parameter coordinates. The distance calculation becomes:
> $d_{\mathbb{L}}(x,x) = \sqrt{K'} \cdot \text{acosh}\left(\frac{-\langle x, x \rangle_{\mathcal{L}}}{K'}\right) = \sqrt{K'} \cdot \text{acosh}\left(\frac{K}{K'}\right)$
> * If $K < K'$, the term $\frac{K}{K'}$ is less than $1$. The $\text{acosh}$ function is undefined for inputs less than $1$, leading to \textit{NaN} values and a model crash.
>
> * If $K > K'$, the calculation does not crash, but the distance is no longer $0$. This incorrect distance creates invalid gradients, which destabilizes training and requires the very strict training regimes (e.g., small learning rates, long warm-ups) that our method seeks to avoid.
>
> This is why previous similar works rely on rigorous training regimes with separate optimizers for the curvature, very small learning rates, and "burn-in" epochs [5,6,7]. Other solutions such as clipping may prevent undefined operations but cannot easily account for the inconsistencies in the second scenario.
>
> &nbsp;
>
> ### “Line 244: ...can you be more specific or concrete in comparing with cutoff/normalization, to prove the principles that you say the new scaling function helps with?"
>
>
>
> Empirically, when using the scaling function vs the sigmoid function and expmap output clipping used in Chen et al. we noticed a few things:
> * We no longer required strict gradient clipping to 1
> * Slightly faster convergence times (~170 epochs vs 200 epochs for vision tasks when not learning the curvature for example)
> * Much less sensitivity to learning rate and weight decay
>
> From a theoretical standpoint, previous work has shown that hard clamping, such as ReLU could lead to "dead neurons" during training due to the 0 gradient and non-smooth training which leads to slower convergence [2]. Other works such as [3], show that hard clamping requires specific clamping thresholds to assure convergence and can fail under the standard noise assumptions.
>
> Alternatively, using other activation functions such as sigmoid or vanilla tanh leads to gradient vanishing in the saturation regions as the derivative of the function approaches 0 on either side [1].
>
> To try and relate the findings to previous literature, we will include a figure of the gradient distribution for parameters per layer during random training epochs. We did this for our method and other clamping and vanilla scaling approaches and show that other methods have a higher proportion of 0 gradients and more gradient spikes. We have not done extensive empirical ablations however (only for vision task) due to the computation cost but we will try to add more to the appendix.
>
> &nbsp;
>
> [1] Nguyen, Anh, et al. "An analysis of state-of-the-art activation functions for supervised deep neural network." _2021 International conference on system science and engineering (ICSSE)_.
> [2] Horuz, Coşku Can, et al. "The Resurrection of the ReLU." _arXiv preprint arXiv:2505.22074_ (2025).
>
> [3] Koloskova, Anastasia, Hadrien Hendrikx, and Sebastian U. Stich. "Revisiting gradient clipping: Stochastic bias and tight convergence guarantees." _International Conference on Machine Learning_. PMLR, 2023.
>
> [4] Mishne, Gal, et al. "The numerical stability of hyperbolic representation learning." _International Conference on Machine Learning_. PMLR, 2023.
>
> [5] Chlenski, Philippe, et al. "Mixed-curvature decision trees and random forests." Forty-second International Conference on Machine Learning (2024).
>
> [6] Skopek, Ondrej, Octavian-Eugen Ganea, and Gary Bécigneul. "Mixed-curvature Variational Autoencoders." International Conference on Learning Representations.
>
> [7] Chlenski, Philippe, et al. "Manify: A Python Library for Learning Non-Euclidean Representations." arXiv preprint arXiv:2503.09576 (2025).

---

> > ### Comment · Reviewer_k73U · 2025-07-31
> > **Thanks for the reply!**
> >
> > I understand the answer to the questions, thank you for taking the time.
> > My appreciation of the paper remains the same, in particular I maintain the same score.

---

> > > ### Author Response · Authors · 2025-08-02
> > >
> > > Thank you very much for your time and your comments! Your comparison suggestion and the graph definitely added insight to the paper.

---

### Official Review · Reviewer_1Enn · 2025-07-02

**Clarity:** 2
**Significance:** 2
**Originality:** 3
**Rating:** 4
**Confidence:** 4

**Summary:**

This paper presents a derivation for Riemannian AdamW that helps increase hyperbolic generalization ability. For improved stability, we introduce a novel fine-tunable hyperbolic scaling approach to constrain hyperbolic embeddings and reduce approximation errors.

**Questions:**

Please see Strengths And Weaknesses

**Ethical Concerns:**

["NO or VERY MINOR ethics concerns only"]

**Final Justification:**

The authors’ rebuttal provides a clear and well-reasoned explanation. However, due to the memory consumption and computational cost of the proposed algorithm, it may be difficult to directly scale it to large models. Overall, the paper presents a solid theoretical exploration. I am willing to adjust my score to Borderline Accept.

**Limitations:**

Yes

**Quality:**

3

**Strengths And Weaknesses:**

​**Strengths**​
- ​**Innovation**:
  - The proposed method sounds interesting.


​**Weaknesses**​
- ​**Insufficient Motivation**:
  - This paper's claim that existing hyperbolic learning approaches remain prone to overfitting, computationally expensive, and susceptible to instability lacks explanatory justification or supporting references, making it unconvincing.
  - The manuscript fails to present a rigorous justification for the effectiveness of the proposed methodology in resolving the aforementioned challenges.
  - Additionally, the proposed work lacks a systematic comparative analysis with state-of-the-art hyperbolic techniques (e.g., HCNN) to explicitly highlight its technical superiority.
- ​**Incomplete Experimental Comparisons**:
  - ​**Outdated Baselines**: Some experiments only compare against outdated methods. Recent hyperbolic methods are omitted.
  - ​**Limited Scope**: Complexity and generalization ability are not involved in the experimental analysis.
  -  **Poor Performance**: The proposed method yields only marginal performance gains, failing to outperform established baselines across multiple evaluation tasks.

---

> ### Author Rebuttal · Authors · 2025-07-31
>
> Thank you for your review.
>
> &nbsp;
>
> ### 1. "This paper's claim that existing hyperbolic learning approaches remain prone to overfitting, computationally expensive, and susceptible to instability lacks explanatory justification or supporting references..."
>
> **Instability and computational costs:** These are widely recognized challenges with hyperbolic machine learning. They are, for example, discussed in the survey papers linked below as [1] and [2], and there are also full papers dedicated exclusively to the study of instability in hyperbolic learning, see [3]. Additionally, we provide concrete examples of instability in the rebuttals to reviewers k73U and 7vca below. We will discuss this background more clearly in the final version of the paper. For computational cost, we demonstrate significant reductions in runtime and memory in Table 3 of our paper, Table 5 in the appendix, and in the response to Reviewer 7vca.
>
> **Overfitting:** We address overfitting by introducing Riemannian AdamW. As discussed in the paper, the improved L2-regularization of AdamW is designed to enhance generalization. The performance gains we demonstrate in data-scarce settings (e.g., EEG classification in Table 2) are consistent with the expected benefits of this stronger regularization.
>
> &nbsp;
>
> ### 2. "the proposed work lacks a systematic comparative analysis with state-of-the-art hyperbolic techniques (e.g., HCNN) to explicitly highlight its technical superiority” and “Outdated Baselines: Some experiments only compare against outdated methods. Recent hyperbolic methods are omitted.”:
>
> We actually compare to the work of Bdeir et al. (both HECNN and HCNN) in Table 3
> and Table 4 in the main text and Table 5 in the appendix. To the best of our knowledge, we compare against the most recent and relevant baselines. If there are other specific methods you believe should be included as baselines, we would be happy to discuss them.
>
> &nbsp;
>
> ### 3. " "Complexity and generalization ability are not involved in the experimental analysis."
>
> We agree that our experimental analysis should explicitly validate our claims of improving computational complexity and generalization. We provide evidence for this in the paper as follows:
>
> **Computational Complexity:** We report a detailed analysis of runtime and memory usage for our method vs. the HECNN baseline in Table 3, showing a ~66% reduction in runtime and a ~48% reduction in VRAM. We provide further analysis in Table 5 of the appendix, and for the VAE models in our response to Reviewer 7vca.
>
> **Generalization Ability:** We address generalization through improved regularization (as mentioned in our first point) and demonstrate its effectiveness with performance gains in data-hungry settings such as EEG classification (Table 2) and hierarchical metric learning (Table 1).
>
> &nbsp;
>
> ### 4. "The proposed method yields only marginal performance gains, failing to outperform established baselines across multiple evaluation tasks."
>
> Our results in Tables 1, 2, 3, and 4 show consistent performance improvements over the established baselines across nearly all evaluated tasks and data sets. Could you please clarify which specific task or baseline you are referring to where our method does not show a clear outperformance?
>
> &nbsp;
>
> [1] Peng, Wei, et al. "Hyperbolic deep neural networks: A survey." _IEEE Transactions on pattern analysis and machine intelligence_ 44.12 (2021): 10023-10044.
> [2] Mettes, Pascal, et al. "Hyperbolic deep learning in computer vision: A survey." _International Journal of Computer Vision_ 132.9 (2024): 3484-3508.
> [3] Mishne, Gal, et al. "The numerical stability of hyperbolic representation learning." _International Conference on Machine Learning_. PMLR, 2023.

---

> > ### Comment · Reviewer_1Enn · 2025-08-07
> >
> > I have read the authors’ responses to both my comments and those from other reviewers. All of my concerns have been fully addressed. I sincerely appreciate the authors’ detailed and patient replies. I will improve my score.

---

### Decision · Program_Chairs · 2025-09-17

**Decision:**

Accept (poster)

**Comment:**

The authors propose a robust curvature-aware optimization framework for hyperbolic deep learning, addressing challenges related to stability, computational efficiency, and overfitting. They introduce Riemannian AdamW, which enhances regularization and generalization in hyperbolic learning, a novel distance-rescaling function that ensures hyperbolic vectors remain within the representative radius of accuracy afforded by numerical precision, and a practical trick that enables efficient CUDA implementations. The efficiency of the proposed method is demonstrated through experiments on several tasks.

The reviewers acknowledge that the paper constitutes a meaningful contribution to hyperbolic learning, noting that:
1. the proposed idea is reasonable and sound
2. the method appears to address key issues present in previous approaches
3. the experimental results convincingly support the claims

Most concerns raised during the review process were addressed in the rebuttal phase. Overall, the paper seems as a significant and impactful contribution, providing a robust and efficient optimization framework for hyperbolic networks.

I therefore recommend acceptance and encourage the authors to consider the feedback and update the paper accordingly.